# High-entropy relaxor ferroelectric ceramics for ultrahigh energy storage

Haonan Peng[1,2,6], Tiantian Wu[3,6], Zhen Liu ®[1] ✉, Zhengqian Fu ®[3], Dong Wang ®[4] ✉, Yanshuang Hao[1], Fangfang Xu ®[3], Genshui Wang ®[1,2,3] ✉ & Junhao Chu[5]

Dielectric ceramic capacitors with ultrahigh power densities are fundamental to modern electrical devices. Nonetheless, the poor energy density confined to the low breakdown strength is a long-standing bottleneck in developing desirable dielectric materials for practical applications. In this instance, we present a high-entropy tungsten bronze-type relaxor ferroelectric achieved through an equimolar-ratio element design, which realizes a giant recoverable energy density of 11.0 J·cm$^{-3}$ and a high efficiency of 81.9%. Moreover, the atomic-scale microstructural study confirms that the excellent comprehensive energy storage performance is attributed to the increased atomic-scale compositional heterogeneity from high configuration entropy, which modulates the relaxor features as well as induces lattice distortion, resulting in reduced polarization hysteresis and enhanced breakdown endurance. This study provides evidence that developing high-entropy relaxor ferroelectric material via equimolar-ratio element design is an effective strategy for achieving ultrahigh energy storage characteristics. Our results also uncover the immense potential of tetragonal tungsten bronze-type materials for advanced energy storage applications.

High-performance energy storage capacitors on the basis of dielectric materials are critically required for advanced high/pulsed power electronic systems. Benefiting from the unique electrostatic energy storage mechanism, dielectric capacitors demonstrate the greatest power density, ultrafast charge/discharge rate, and long-life work time. Consequently, it has been utilized in a wide variety of high-tech industries, including medical devices, military equipment, and hybrid electric vehicles[1–4]. Nevertheless, in comparison to electrochemical capacitors and batteries, the inferior energy storage capability of current candidate dielectric ceramics impedes their wider application and developments toward miniaturization, lightweight, and cost

reduction[5–7]. Consequently, exploring novel ceramic compositions that possess a high energy storage density is essential for pulsed power system applications. In accordance with the theoretical calculation formula of electrostatic energy storage:

$$W_{\mathrm{rec}} = \int_{P_{\mathrm{r}}}^{P_{\mathrm{m}}} E dP \qquad (1)$$

a large maximum polarization ($P_{\mathrm{m}}$), a small remnant polarization ($P_{\mathrm{r}}$), and a high breakdown electric field ($E_{\mathrm{b}}$) is essential for attaining a substantial density of recoverable energy storage ($W_{\mathrm{rec}}$)[8,9].

[1]Key Laboratory of Inorganic Functional Materials and Devices, Shanghai Institute of Ceramics, Chinese Academy of Sciences, Shanghai 200050, China. [2]Center of Materials Science and Optoelectronics Engineering, University of Chinese Academy of Sciences, Beijing 100049, China. [3]State Key Laboratory of High Performance Ceramics and Superfine Microstructures, Shanghai Institute of Ceramics, Chinese Academy of Sciences, Shanghai 200050, China. [4]Frontier Institute of Science and Technology and State Key Laboratory for Mechanical Behavior of Materials, Xi'an Jiaotong University, 710049 Xi'an, Shaanxi, China. [5]State Key Laboratory of Infrared Physics, Shanghai Institute of Technical Physics, Chinese Academy of Sciences, Shanghai 200083, China. [6]These authors contributed equally: Haonan Peng, Tiantian Wu. ✉e-mail: zhenliu@mail.sic.ac.cn; wang_dong1223@mail.xjtu.edu.cn; genshuiwang@mail.sic.ac.cn

Unfortunately, due to the inherent feature of typical dielectric materials, i.e., large $P_r$ for ferroelectrics (FEs), low $P_m$ for linear dielectrics (LDs), and large hysteresis for antiferroelectrics, it is challenging to attain optimal energy storage efficiency and density in conventional dielectric materials[10,11]. To optimize energy-storage performance, polar nanoregions (PNRs) with low energy barriers for polarization switching are typically constructed through relaxor design, resulting in slim $P$-$E$ loops with high $P_m$ and low $P_r$[12–14]. Hence, to attain superior energy storage performance in advanced dielectric ceramics, relaxor design has emerged as the most promising approach[15–17].

Enhanced compositional inhomogeneity typically induces local structure disorder and polar fluctuations, which interfere with the long-range ferroelectric order and stimulate enhanced relaxor behavior[18,19]. High-entropy materials allow five or more ions with distinct radii and valences to occupy the equivalent lattice sites in an equimolar or near-equimolar ratio, consequently enhancing the heterogeneity of composition at the atomic level[20]. Meanwhile, taking advantage of the unique entropy-dominated phase stabilization, lattice distortions, sluggish diffusion, as well as property synergies of multiple components[21], high-entropy ceramics produce optimized dielectric parameters, including high permittivity[22] and low dielectric loss[23]. Accordingly, developing high-entropy ceramics is anticipated to serve as an effective method of relaxor design as well as enhance energy storage performance[24]. Tetragonal tungsten bronze (TTB) structures, with the general formula $A1_2A2_4C_4B_{10}O_{30}$, attract extensive attention due to their complex structure and abundant dielectricity and ferroelectricity, in spite of their relatively low polarization strength and low $E_b$ caused by abnormal grain growth[25–27]. TTB contains 12-fold-coordinated quadrilateral A1 sites, 15-fold-coordinated pentagonal A2 sites, and 9-fold-coordinated triangular C sites (typically empty) on the basis of the networks of corner-sharing $BO_6$ octahedra[28,29]. The existence of two A-sites, A1 and A2, with different coordination, in TTB structures would bring about multiple possibilities for configuration entropy regulation, resulting in higher atomic disorder and compositional heterogeneity. Unfilled TTB, which attain electrical neutrality by filling 5/6 of the A-sites while leaving 1/6 A-site vacancies, have the potential to generate further entropy increase. This is primarily attributed to the disorderly distribution of cations and vacancies[30–32]. Consequently, the TTB structures are an ideal choice for designing high-entropy ferroelectric materials with highly disordered atomic distribution and enhanced relaxation, and promisingly extend the area available for optimizing the performance of energy storage.

Here, with the consideration of achieving high polarization and diversifying ion valence states and radii, the common TTB A-site ions $Sr^{2+}$ and $Ba^{2+}$ from the classical unfilled TTB $(Sr_{0.5}Ba_{0.5})Nb_2O_6$ ceramic, $Pb^{2+}$ with high polarizability, and another two heterovalent ions $La^{3+}$ and $Na^+$, are selected as the high-entropy components. By introducing the above equimolar-ratio elements with distinct valence and radii at A sites, we effectively designed and fabricated high-entropy TTB-structured ceramics with a composition of $(Sr_{0.2}Ba_{0.2}Pb_{0.2}La_{0.2}Na_{0.2})Nb_2O_6$ (SBPLNN). Besides, through an atomic-scale structural analysis utilizing aberration-corrected scanning transmission electron microscopy (STEM), an increased compositional heterogeneity along with a strong nonperiodic lattice distortion was observed. The former induces PNRs and modulates the relaxor properties, thus defining the reduced polarization switching hysteresis. The latter leads to grain refining, suppressed leakage current density, as well as superior electrical homogeneity, which finally substantially enhances the breakdown strength. Consequently, our designed high-entropy ceramics simultaneously realize an ultrahigh $W_{rec}$ of 11.0 J·cm$^{-3}$ and a high $\eta$ of 81.9% under a high electric field of ~ 753 kV·cm$^{-1}$, in addition to possessing exceptional fatigue, temperature, frequency stability, and charge-discharge performance. Except for that, the excellent energy storage performance of SBPLNN ceramics extends beyond recently reported state-of-the-art dielectric ceramics and constitutes a substantial advancement over current TTB ceramics. The present study introduces the equimolar-ratio element high-entropy strategy as a universal, practical, and efficient method for developing dielectric materials of the next generation that exhibit exceptionally high energy storage capabilities.

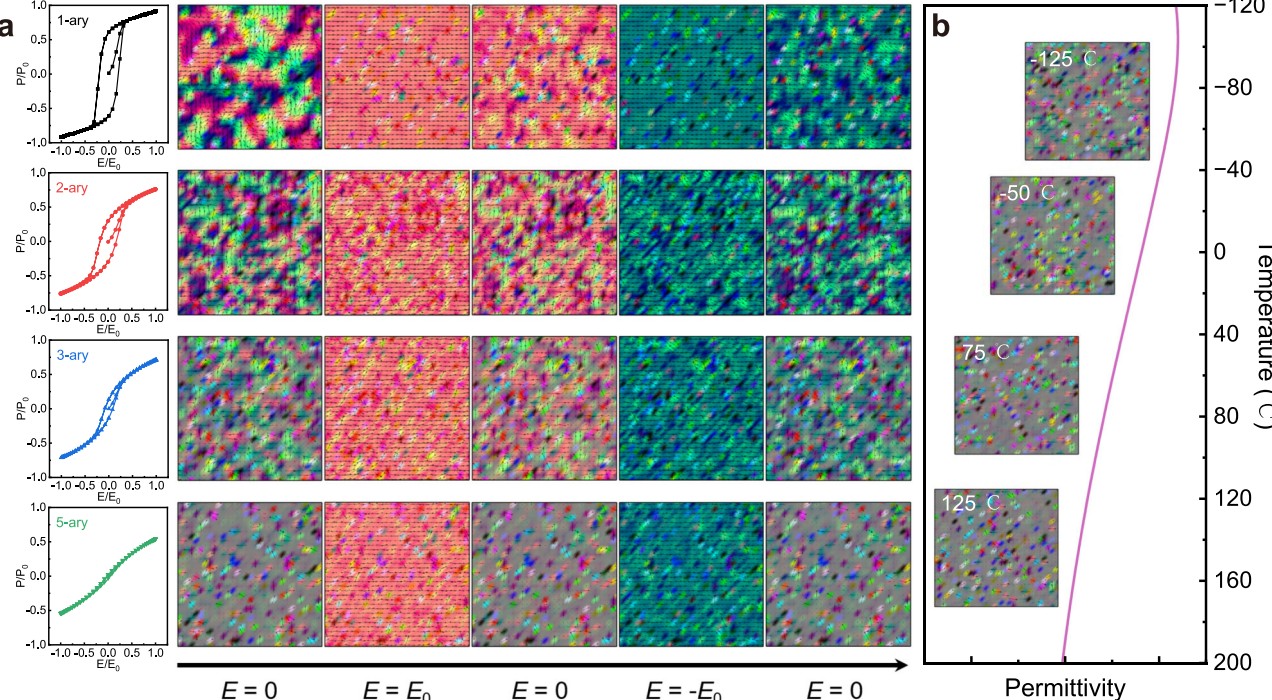

**Fig. 1 | Phase-field simulations results. a** Simulation results of $P$-$E$ hysteresis loops and domain structure evolution with an increasing number of A-site element species during the application and removal of electric field. **b** Simulation results of the domain structure of 5-ary high-entropy ceramics during a wide temperature range. The gray color represents the paraelectric phase, the other colors distinguish the ferroelectric domains with different orientations (arrows).

## Results and discussion

### Phase-field simulations of domain structures and *P-E* loops

In the first place, we utilized phase-field simulations to guide the design of high-entropy ceramics. We subsequently investigated the evolution of domain structures and *P-E* loops as the species of A-site element increase from 1-ary to 2-ary, 3-ary, and finally 5-ary, as depicted in Fig. 1a. In this study, we conduct two-dimensional phase field simulations using a 1-ary (single element) system as the matrix and implement guest doping with various elements. The 1-ary system exhibits characteristics of a traditional ferroelectric with large ferro-electric domains. The polarization is nearly entirely in accordance with the direction of the external electric field, and the initial state cannot be fully restored once the external electric field has been removed, maintaining a large degree of polarization distribution. As the species of A-site element increase, the equimolar foreign atoms with unmat-ched atomic size, mass, and electronegativity act as point defects of identical proportion, which induce the local compositional inhomo-geneity and thus destroy the ferroelectric long-range order into PNRs and enhance the relaxor behavior[33,34]. Consequently, domain sizes gradually decrease as the number of element species increases. Moreover, the polarization directions progressively become dis-ordered, thus delaying the polarization saturation process. Meanwhile, the smaller domains show higher activity, facilitating a rapid return to the initial state subsequent to the cessation of the electric field[35]. These microstructural changes contribute to *P-E* loops becoming gradually slimmer, especially in 5-ary high-entropy ceramics. Such an enhanced relaxor behavior generally exhibits a diffused phase transition with a broad temperature range of dielectric relaxation and a relatively flat dielectric spectrum. Subsequently, we calculated the domain structure evolution of 5-ary high-entropy ceramics across a broad temperature range. As illustrated in Fig. 1b, both the polarization distribution state and domain size are insensitive to temperature variation, implying the temperature-insensitive macroscopic polarization-related property. Thus, the energy storage potential and temperature stability of strong relaxors in 5-ary high-entropy ceramics are predicted by the phase field simulation results.

### Energy storage properties, stability, and charge/discharge performance

Directed by the phase field simulation outcomes, we designed and fabricated $(Sr_{0.2}Ba_{0.2}Pb_{0.2}La_{0.2}Na_{0.2})Nb_2O_6$ high-entropy ceramics with an equimolar ratio by introducing five A-site elements, which are commonly adopted in TTB-based materials. The X-ray Rietveld refinement (Fig. S1) indicates that SBPLNN high-entropy ceramics possess a pure tungsten bronze structure with a *P4bm* tetragonal space group. This entropy-dominated phase stabilization effect can be comprehended by the reduced Gibbs free energy

$$\Delta G_{mix} = \Delta H_{mix} - T\Delta S_{mix} \qquad (2)$$

where $G_{mix}$, $H_{mix}$, and $S_{mix}$ are the mixed free energy, enthalpy, as well as entropy, respectively, and $T$ indicates the kelvin temperature). The $\Delta S_{mix}$ rises to a high entropy of 1.61R with five equimolar elements ($Sr^{2+}$, $Ba^{2+}$, $Pb^{2+}$, $La^{3+}$, and $Na^+$) introduced to the A sites, resulting in a reduction in the $\Delta G_{mix}$. This change in free energy manifests as an entropy-driven structural stabilization effect, facilitating the formation of a stable, complex single-phase structure[36]. On the other hand, typical TTB-structured ceramics generally produce anisometric rod grains with large grain sizes, due to the lower surface energy of the (001) facet induced faster grain growth than other directions, which restricts the achievement of a high $E_b$[27]. Conversely, SBPLNN illustrates a dense microstructure accompanied by refined and equiaxed grain sizes of ~1.57 μm (Fig. S2a). In the meantime, the energy dispersive X-ray spectroscopy (EDS) mapping of a region of 40 × 30 μm (Fig. S2b) demonstrates a uniform element distribution in the absence of element segregation. The above experimental results verify the successful fabrication of single-phase high-entropy tungsten bronze ferroelectric ceramics.

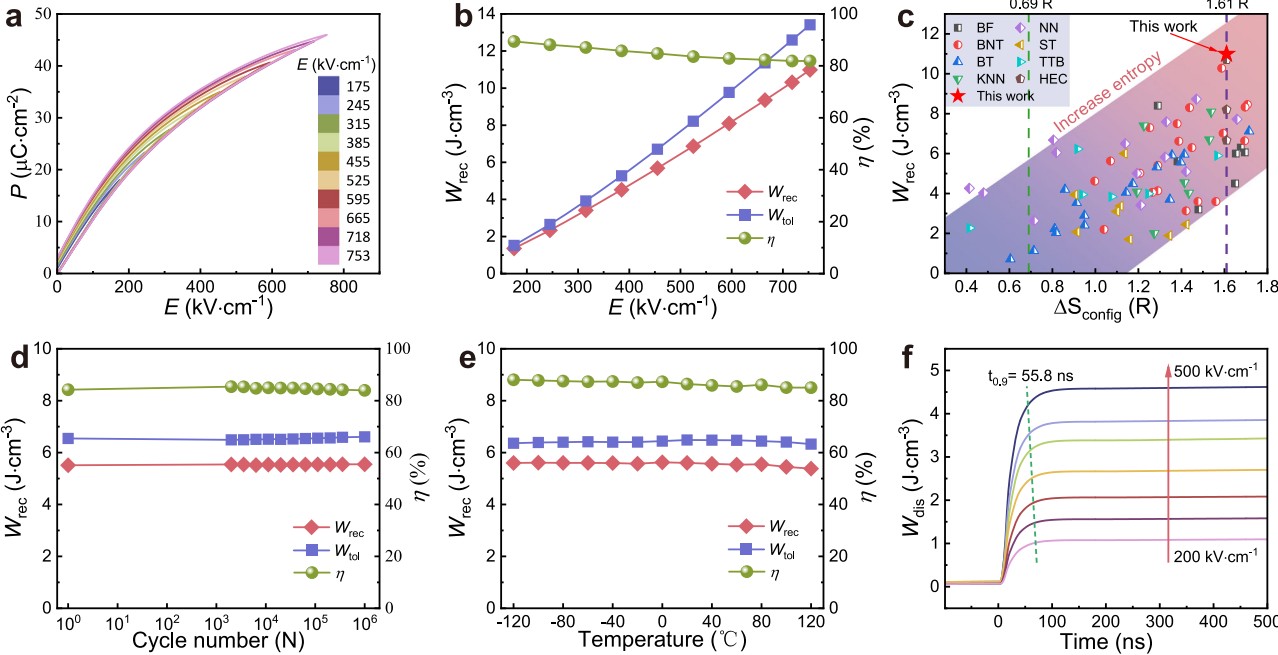

**Fig. 2 | Energy storage and charge/discharge performance of SBPLNN ceramics.** **a** *P-E* hysteresis loops till the maximum test field of SBPLNN ceramics. **b** $W_{tol}$, $W_{rec}$, and $\eta$ as a function of an electric field. **c** Comparison of the $W_{rec}$ of reported dielectric ceramics with various $\Delta S_{config}$ calculated by this work. **d** $W_{tol}$, $W_{rec}$, and $\eta$ as a function of cycle number under 470 kV·cm⁻¹. **e** $W_{tol}$, $W_{rec}$, and $\eta$ as a function of temperature under 470 kV·cm⁻¹. **f** $W_{dis}$ as a function of time under different electric fields ($R$ = 300 Ω).

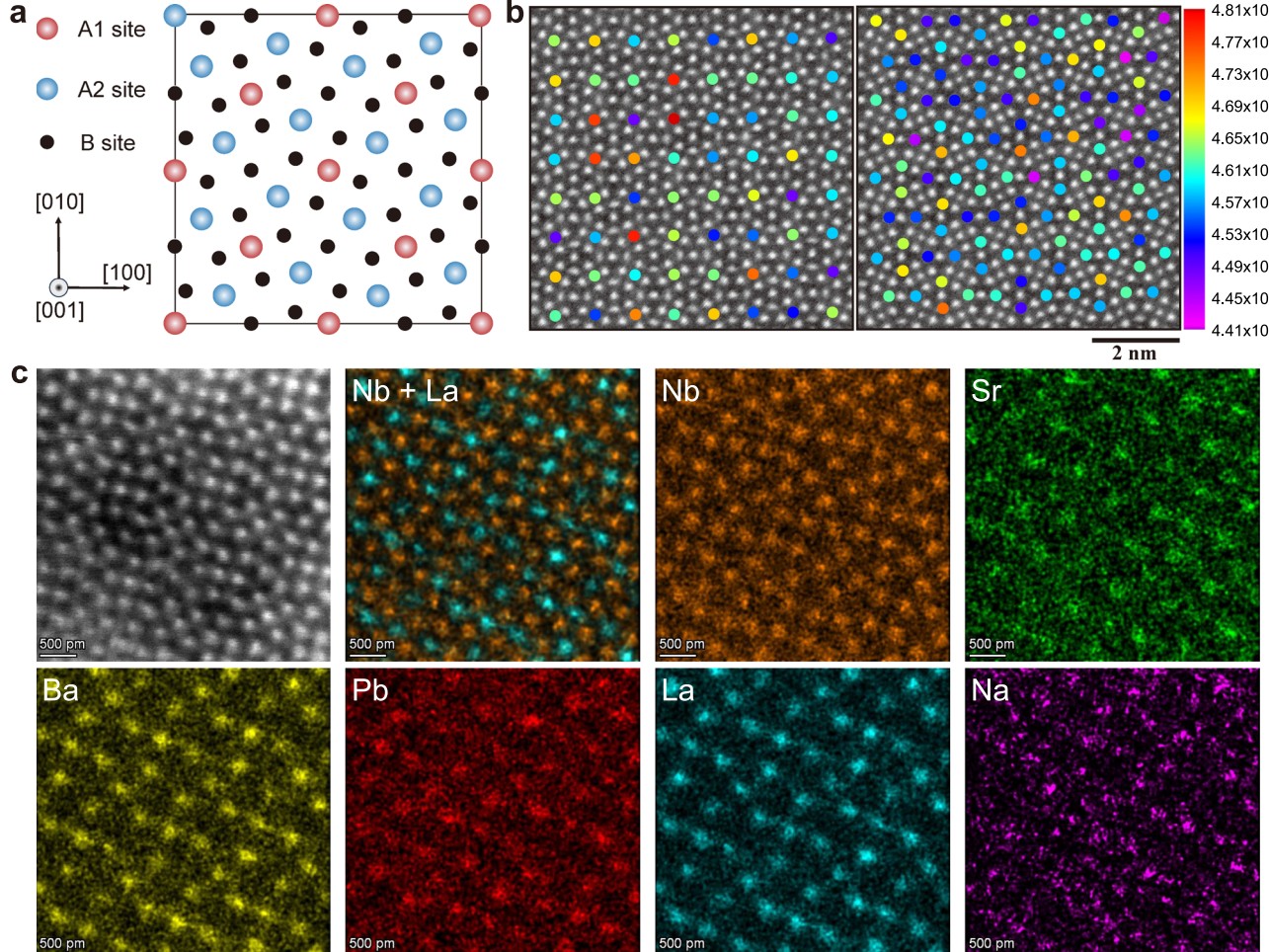

**Fig. 3 | Local compositional inhomogeneity of high-entropy ceramics.**
**a** Schematic lattice structure of the tetragonal tungsten bronze projected along the
[001] axis. **b** The atomic column intensities of the A1 and A2 sublattices for SBPLNN

ceramics. The color of the circles indicates the intensity of each atom column.
**c** HAADF images along [001] zone axes and the atomically resolved elemental
mappings of SBPLNN ceramics, including distribution of Sr, Ba, Pb, La, Na, and Nb.

We measured the dielectric properties of SBPLNN ceramics over a temperature range of −120 °C to 200 °C and a range of frequencies from 10 Hz to 1 MHz (Fig. S3a). The $T_m$ of SBPLNN increases with increasing frequency, and the diffuse dielectric peaks reveal a large $\Delta T$ ($T_{m,1MHz}$-$T_{m,10Hz}$) value of 60.5 °C. In accordance with the modified Curie−Weiss law

$$\frac{1}{\varepsilon} - \frac{1}{\varepsilon_m} = \frac{(T - T_m)^\gamma}{C} \qquad (3)$$

the high diffuseness exponent $\gamma$ (1.58) (Fig. S3b) substantiates the improved relaxor performance of our SBPLNN ceramics. Meanwhile, the dielectric loss of SBPLNN ceramics is greatly suppressed, which drops down to ~0.0014 at room temperature and 1 kHz. Hence, P-E hysteresis loops maintain a slender feature and present a high $P_{max}$ of ~45.9 µC·cm⁻² accompanied by a low $P_r$ of ~ 2.8 µC·cm⁻² under the maximum test field, thus demonstrating a reduced hysteresis loss (Fig. 2a). Moreover, SBPLNN high-entropy ceramics can also with-stand an ultrahigh electric field of ~ 753 kV·cm⁻¹. Consequently, a giant $W_{rec}$ of 11.0 J·cm⁻³ and a high $\eta$ of 81.9% are acquired simultaneously. As per the principles of classical superparaelectric theory, the $T_m$ of SBPLNN ceramics decreases to below zero, shifting the superparaelectric region to near room temperature. In this case, enhanced relaxor behavior coupled with ultrasmall PNRs appears, which diminishes energy loss under high electric fields and ensures

the significant advancement of $W_{rec}$ and $\eta$ simultaneously[4]. Fig. S4 summarizes the $W_{rec}$ and $E$ of current state-of-the-art energy storage ceramics. The SBPLNN ceramics present a record-high $W_{rec}$ among existing TTB-structured ceramics. This unprecedented breakthrough in $W_{rec}$ has likewise exceeded the recently developed advanced perovskite ceramics, including NaNbO₃-based, (Bi, Na)TiO₃-based, AgNbO₃-based, BaTiO₃-based, (K, Na)NbO₃, etc. Additionally, we also calculate the $\Delta S_{config}$ of the reported advanced bulk ceramics and perform a comparison of their $W_{rec}$ as well as the $\Delta S_{config}$ in Fig. 2c. Regarding the whole, the $W_{rec}$ indicates an increment trend with the increase of configurational entropy, and our SBPLNN ceramics occupy the highest position among these reported materials with different entropy values. Moreover, it is essential to note that recently documented high-entropy strategies for dielectric materials with high energy storage capacity are predominantly developed using a non-equal molar ratio approach, which heavily depends on intricate and irregular component design and element ratio regulation[37-39]. In contrast, by simply introducing equimolar-ratio atoms, we efficiently realize superior energy storage performance in SBPLNN high-entropy ceramics. The outcomes of our study demon-strate that the equimolar ratio high-entropy design is a practical and efficient method for developing advanced pulse power dielectric materials.

It is substantial to maintain the stability of energy storage ceramics for a reliable practical application in extreme conditions.

The fatigue resistance of SBPLNN ceramics is robust up to $10^6$ cycles under 470 kV·cm$^{-1}$ (Figs. 2d and S5a). The high $W_{rec}$ exhibits no obvious deterioration with a slight fluctuation from 5.51 J·cm$^{-3}$ to 5.55 J·cm$^{-3}$, and the $\eta$ marginally declines from 85.41% to 83.98%. Simultaneously, the excellent temperature stability of SBPLNN ceramics at −120 °C ~ 120 °C is illustrated in Figs. 2e and S5b. During the heating process, enhanced random electric fields typically alleviate the response of PNRs, resulting in a decrease in polarization strength[16]. While, throughout the broad testing temperature range, the narrow feature of P-E loops stays unchanged, resulting in a stable high $W_{rec}$ ~ 5.50 ± 0.12 J·cm$^{-3}$ and $\eta$ ~ 86.54 ± 1.51%. Temperature-dependent XRD and Raman spectroscopy are tested to investigate the thermal structural evolution. The positions and numbers of diffraction peaks remain unaltered with a temperature change from −160 °C to 290 °C (Fig. S6), indicating the structure stability of our entropy-stabilized SBPLNN ceramics. As depicted in Fig. S7, across the broad temperature range of −160 °C–300 °C, every Raman mode exhibits a diffused and broadened characteristic, which signifies the potent relaxor nature of SBPLNN ceramics. External vibration modes (below 200 cm$^{-1}$) indicate strong scattering intensity and complex peak shapes, which are in association with the distinct charges and radii between A site atoms of SBPLNN ceramics. This cation disorder contributes to the local random electric fields and induces the formation of PNRs in a wide temperature range[40,41]. As seen in Fig. S8, we also employed the phase field simulation to predict the temperature-dependent polarization response. As the temperature increases from −120 to 120 °C, the size of PNRs keeps overall stable while slightly decreasing, making $P_m$ slightly vary from 36.43 μC·cm$^{-2}$ to 31.94 μC·cm$^{-2}$. During the whole range, PNRs maintain high activity and demonstrate facilitated polarization rotation, thus ensuring slim P-E loops with high $P_m$ and low $P_r$. Additionally, SBPLNN ceramics also demonstrate excellent frequency stability (Fig. S5c, d). The P-E loops maintain slimness with nearly unchanged $P_{max}$ (34.50 ± 0.68 μC·cm$^{-2}$) and $P_r$ (1.80 ± 0.29 μC·cm$^{-2}$) from 10–250 Hz, resulting in a stable $W_{rec}$ ~ 5.55 ± 0.06 J·cm$^{-3}$ and $\eta$ ~ 88.31 ± 1.93%. The overdamped (load resistance, $R$ = 300 Ω) discharge behavior of SBPLNN ceramics is illustrated in Figs. 2f and S9c. As the applied electric field rises from 200 kV·cm$^{-1}$ to 500 kV·cm$^{-1}$, the current peak increases from 1.96 A to 4.41 A. The discharge energy density ($W_{dis}$) is calculated by the following equation:

$$W_{dis} = \frac{R \int i^2(t) dt}{V} \qquad (4)$$

where $V$ represents the sample volume. A high $W_{dis}$ of 4.58 J·cm$^{-3}$ is acquired, and the time required for releasing 90% of the discharged energy ($t_{0.9}$) is determined as 55.8 ns, revealing an ultra-fast discharge speed[42]. In Fig. S9a, b, the underdamped charge/discharge performance is also assessed. The current waveform is smooth and stable, with peak current ($I_{max}$) achieving 15.83 A at 500 kV·cm$^{-1}$. According to the formulas

$$C_D = \frac{I_{max}}{S} \qquad (5)$$

$$P_D = \frac{E I_{max}}{2S} \qquad (6)$$

where $S$ denotes sample electrode area, the calculated current density ($C_D$) and power density ($P_D$) attain a substantial value of 2488.84 A·cm$^{-2}$ and 622.21 MW·cm$^{-3}$, respectively[43]. It is evident that SBPLNN ceramics demonstrate substantial improvements in energy storage performance, including ultrahigh energy density, high energy efficiency, superior frequency/temperature/fatigue stability, as well as discharging performance. Consequently, the great potential of SBPLNN

ceramics for practical applications as high-power pulse capacitors is exceedingly emphasized.

## The local compositional inhomogeneity and enhanced relaxor behavior

For the purpose of investigating the impacts of high configuration entropy on the compositional inhomogeneity of SBPLNN ceramics, we acquired the atom image along the [001] zone axis by aberration-corrected STEM with high-angle angular dark-field (HAADF) imaging. Figure 3a illustrates the schematic lattice of the TTB structure when viewed along the [001] axis. Within this structure, A1 and A2 atoms occupy quadrilateral and pentagonal sites, respectively, which were formed by B-site atoms. The intensity of an atom column in the HAADF-STEM is strongly dependent on the atomic number ($Z$) and corresponding sublattice distortion[44]. As depicted in Fig. 3b, the intensity of the atomic column of A1 and A2 sublattices is comparatively diffuse, which displays a slight and random space-dependent fluctuation. Moreover, the average atomic column strength of A1 sublattices differs from that of A2 sublattices by less than 2%, being ~4.62 × 10$^5$ and ~4.58 × 10$^5$ for A1 and A2, respectively. Since the atomic numbers of five A-site elements are quite distinct (Sr$^{2+}$ ($Z$ = 38), Ba$^{2+}$ ($Z$ = 56), Pb$^{2+}$ ($Z$ = 82), La$^{3+}$ ($Z$ = 57), and Na$^+$ ($Z$ = 11)), this nearly identical average atomic column strength of two A-sites demonstrates that the atomic distribution of our high-entropy ceramics is disordered overall.

Nonetheless, the TTB structures usually possess unique inherent atomic A1- or A2-site selectivity according to the ionic radius. For instance, in the lattice of typical TTB-structured Ba$_2$NaNb$_5$O$_{15}$, all the larger Ba$^{2+}$ (1.61 Å) occupy A2 sites, whereas the smaller Na$^+$ (1.39 Å) only distribute in A1 sites[45]. Given the large difference in ionic radius (Sr$^{2+}$(1.44 Å, 12CN), Ba$^{2+}$ (1.61 Å, 12CN), Pb$^{2+}$(1.49 Å, 12CN), La$^{3+}$ (1.36 Å, 12CN), and Na$^+$(1.39 Å, 12CN)), should the ideal atomic site selectivity have been obeyed in SBPLNN ceramics, the atomic column strengths of sites A1 and A2 will diverge significantly, which contradicts with the above-tested results. Subsequently, we performed the atomic-resolved EDS mapping to quantitatively analyze the individual element distribution of SBPLNN high-entropy ceramics. The EDS mapping indicates that each element enters the quadrilateral A1 and pentagonal A2 sublattice simultaneously, irrespective of the value of the ionic radius (Figs. 3c and S10). We subsequently quantitatively extracted the distribution of each element in two A-site sublattices according to its EDS signal intensity, while the inferior signal strength and contrast of Na$^+$ were challenging to analyze quantitatively. Consequently, we qualitatively calculate the distribution of Na$^+$ on the basis of the calculation results of the other four elements and atomic column intensity. As listed in Table S1, the calculated ratio of each element occupying the A1 and A2 sublattices approximates the ideal value (0.5). The ionic radius is commonly the intrinsic factor that governs the atomic site-selective distribution in tungsten bronze structures[28]. Nevertheless, in the circumstances of five elements with considerable radius differences in our case, the high entropy-driven slow diffusion effect suppresses atom movement, thus achieving disordered element distribution. Our results confirm that high-entropy effects break the inherent site selectivity of TTB structures and confirm the enhanced local compositional inhomogeneity of SBPLNN high-entropy ceramics.

The random distribution of site atoms is contemplated to be the structural origin of the enhanced relaxor behavior of TTB ceramics[26,31,46], which induces structural disorder and disturbances of B-site polarity displacement, thus inducing a phase transition from normal ferroelectric to relaxor ferroelectric by impeding the stabilization of long-range ferroelectric ordering[30,47,48]. As for SBPLNN high-entropy ceramics, five ions with large radius differences occupying the two coordination sites (A1 and A2) yield enhanced local compositional inhomogeneity, which results in local structure disorder and polar

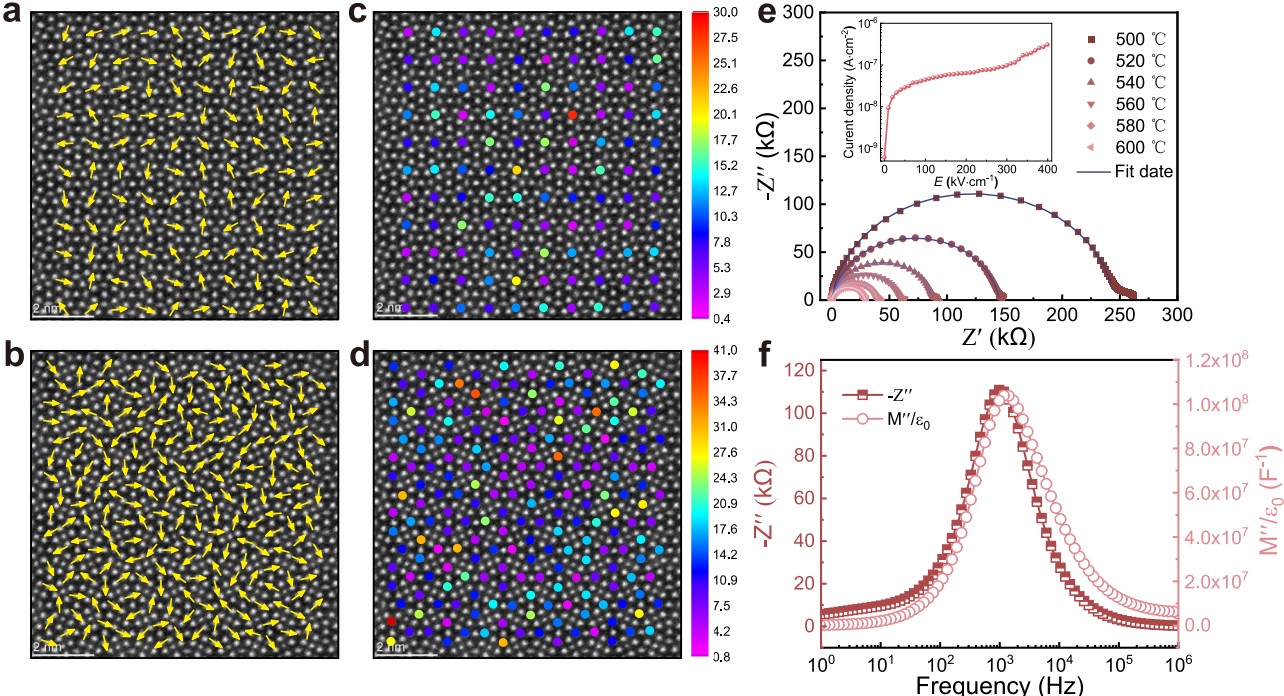

**Fig. 4 | Lattice distortion in the high-entropy ceramics.** Atomic displacement directions of (**a**) A1 and (**b**) A2 sites atoms with respect to the corner atoms (Nb). The mappings of atomic displacement magnitude of (**c**) A1 and (**d**) A2 sites atoms with respect to the corner atoms (Nb). **e** $Z^*$ plots of SBPLNN ceramics measured from 500 °C to 600 °C. The inset shows the leakage current density. **f** Spectroscopic plots of $Z''$ and $M''$ spectra at 500 °C for the SBPLNN ceramics.

fluctuations and ultimately modulates the relaxor features[49]. The long-range ferroelectric order is disrupted and PNRs are generated due to the significant local composition fluctuations. It can be seen that SBPLNN high-entropy ceramics show ultrasmall spot-like nanodomains (Fig. S11a), in contrast to the large-size ferroelectric domain of $Sr_{0.5}Ba_{0.5}Nb_2O_5$ (SBN)[50–52]. Moreover, SBN ceramics display evident piezoelectric signals after 15 V voltage poling, which can be maintained after 15 min relaxation duration[27]. On the contrary, SBPLNN ceramics present a much weaker piezoelectric response under a higher voltage of 40 V, illustrating a delayed polarization reversion behavior upon electric field application. The piezoelectric signals completely disappear after 15 min, revealing enhanced domain mobility (Fig. S11b–d). The weak contrast observed in the STEM results of SBPLNN ceramics (Fig. S12) again supports the formation of ultrasmall PNRs in our high-entropy ceramics. The formation of high-activity PNRs brings reduced polarization switching hysteresis with negligible remnant polarization, which promises high $\eta$. Simultaneously, the enhancement of $\eta$ reduces the joule heat, thus lowering the risk of thermal breakdown[3,53].

### The lattice distortion and improved electric strength
The local compositional inhomogeneity likewise generates a large influence on the lattice structure. To be precise, neighboring atoms with different radii, masses, and electronegativity cause random deviation from their perfect site, which leads to high entropy ceramics experiencing severe structural disorder[21,54]. Thus, we quantitatively analyzed the lattice structure of SBPLNN high-entropy ceramics by calculating the atomic displacement directions and magnitude of A1 and A2 site atoms concerning the corner atoms (Nb), from the atom image along the [001] zone axis. In general, the atomic displacement directions of the A1 and A2 sites exhibit a significant degree of disorder (Fig. 4a, b). As indicated by the statistical distribution bar chart (Fig. S13a, c), displacement directions randomly distribute from −180° to 180°, which indicates a non-periodic lattice distortion in SBPLNN high-entropy ceramics. Meanwhile, as the mappings and statistical

distributions of magnitude illustrate (Figs. 4c, d and S13b, d), the displacement magnitude of A1 and A2 site atoms fluctuates evidently and distributes randomly. The atomic displacements of A1 site atoms range from 0 to 30 pm, with an average value of 8.8 pm. Due to the larger size of the pentagonal A2 site than the quadrilateral A1 site, the atomic displacements of A2 site atoms vary between 0 and 41 pm, with a larger average value of 13.1 pm. For ideal TTB-structured lattices, like $Ba_2NaNb_5O_{15}$, $Ba^{2+}$, and $Na^+$ located in A2 and A1 sites, respectively, generating a distortion-free perfect periodic high-symmetric lattice. The large and random atomic displacements in our results indicate the existence of strong non-periodic lattice distortion of SBPLNN high-entropy ceramics.

The influence of such an atomic disorder and lattice distortion on microstructure would be grain refining[24,55]. As indicated in Fig. S2a, SBPLNN ceramics demonstrate an average grain size of 1.57 μm with an equiaxed feature. The direct effect of grain refinement is an increase in grain boundary concentration, which obstructs the transport of charge carriers significantly by utilizing depletion layers at grain boundaries[56,57]. Moreover, strong lattice distortion generally increases the probability of collision and scattering of electrons with twisted lattice atoms[55,58]. Consequently, the strong lattice distortion enables SBPLNN ceramics to have a greater resistivity. The impedance spectroscopy (Fig. 4e) indicates the resistance is about 1.7–5.0 times higher than other classical TTB compositions (Table S2). And the fitting results (Fig. S14) indicate a large grain boundary conduction activation energy ($E_{a,gb}$) of -1.32 eV, which increases the energy barrier at which oxygen vacancies can engage in conduction[59]. Hence, the leakage current density is suppressed to $-2.9 \times 10^{-8}$ A·cm$^{-2}$ at 50 kV·cm$^{-1}$ and maintains a value lower than $-9.7 \times 10^{-8}$ A·cm$^{-2}$ until a high $E$ of 300 kV·cm$^{-1}$ (inset of Fig. 4e), which is 1–2 orders of magnitude lower than the value of other reported TTB and perovskite structure compositions tested under a low electric field (Table S3). The extremely low leakage current density is an important contributor to the greatly suppressed electric breakdown. On the other hand, electrical

inhomogeneity generally leads to short-circuit conduction pathways, leading to finite electric strength[60]. We, therefore, investigate the electrical heterogeneity by contrast spectroscopic plots of $Z''$ and $M''$, which denote grain (electroactive regions) and grain boundary (resistive electroactive region) responses[61], respectively. A single $Z''$ and $M''$ Debye peaks appear at nearly the same frequency (Fig. 4f), illustrating the superior electrical homogeneity of SBPLNN high-entropy ceramics. These results confirm that the large non-periodic lattice distortion brings about a refined grain effect accompanied by higher resistivity, as well as superior electrical homogeneity and low leakage current density, which ultimately contribute to an ultrahigh breakdown strength ~753 kV·cm$^{-1}$ (Fig. S15), thus guaranteeing the ultrahigh-energy storage performance of SBPLNN high-entropy ceramics[62].

In summary, we presented a novel equimolar-ratio high-entropy $(Sr_{0.2}Ba_{0.2}Pb_{0.2}La_{0.2}Na_{0.2})Nb_2O_6$ tungsten bronze structured relaxor ferroelectric, providing an ultra-high $W_{rec}$ of 11.0 J·cm$^{-3}$ and a high efficiency ($\eta$) of 81.9%. Moreover, we conducted systematic microstructural investigations via HAADF-STEM, which illustrated that high-entropy effects break the inherent atomic site selectivity of TTB structures, thereby enhancing the atomic-scale compositional heterogeneity and inducing intense non-periodic lattice distortion. The local compositional inhomogeneity causes structure disorder and polar fluctuations, induces the formation of PNRs, and greatly reduces polarization hysteresis. Meanwhile, benefiting from the high-entropy-induced intense non-periodic lattice distortion, a grain refinement effect accompanied by low leakage current density and superior electrical homogeneity is observed, ultimately resulting in an ultrahigh breakdown strength. Concurrently, SBPLNN ceramics present a broad usage temperature range of −120 °C ~ 120 °C, enduring fatigue stability of up to $10^6$ cycles, and a broad frequency range of 10 Hz ~ 250 Hz, with negligible fluctuation of less than 2.2% in energy density. This work will open a new era by underscoring a convenient, effective, and universal strategy for achieving superior comprehensive energy storage performance via equimolar-ratio high-entropy design to better meet advanced energy storage application requirements.

## Methods

### Phase field method

In phase-field simulations, a single crystal undergoing a ferroelectric transition from Cubic (C) to Tetragonal (T) phase has been introduced. The total free energy can be described as[63]

$$F = \int_V (f_{bulk} + f_{grad\text{ient}})dV + \int_V f_{couple}dV + \int_V (f_{elas} + f_{elec})dV \quad (7)$$

the bulk free energy density $f_{bulk}$ can be described as follows[64,65]

$$
\begin{aligned}
f_{bulk} &= \alpha_1(\mathbf{p}^2) - \alpha_{11}(\mathbf{p}^2)^2 + \alpha_{111}(\mathbf{p}^2)^3 + \alpha_{12}(\sum_{i,j,i\neq j}^{3} P_i^2 P_j^2) \\
&\quad + \alpha_{112}(\sum_{i,j,i\neq j}^{3} P_i^4 P_j^2) + \alpha_{113}(P_1^2 P_2^2 P_3^2) \\
&= \alpha_1(P_1^2 + P_2^2 + P_3^2) - \alpha_{11}(P_1^2 + P_2^2 + P_3^2)^2 + \alpha_{111}(P_1^2 + P_2^2 + P_3^2)^3 \\
&\quad + \alpha_{12}(P_1^2 P_2^2 + P_2^2 P_3^2 + P_1^2 P_3^2) + \alpha_{112}(P_1^4 P_2^2 + P_2^4 P_3^2 + P_1^4 P_3^2 \\
&\quad + P_1^2 P_2^4 + P_2^2 P_3^4 + P_1^2 P_3^4) + \alpha_{113}(P_1^2 P_2^2 P_3^2)
\end{aligned}
\quad (8)
$$

where $\alpha_{ij}$ is the coefficient and depends on doping concentration and temperature. The gradient energy $f_{grad}$, is written in terms of $P$ as follows:

$$
\begin{aligned}
f_{gradient} = \frac{1}{2}G_{11}&((P_{1,1})^2 + (P_{1,2})^2 + (P_{1,3})^2 + (P_{2,1})^2 + (P_{2,2})^2 + (P_{2,3})^2 \\
&+ (P_{3,1})^2 + (P_{3,2})^2 + (P_{3,3})^2)
\end{aligned}
\quad (9)
$$

where $G_{11} = 1.5 \times 10^{-10}$ Jm$^3$C$^{-2}$. The local stress field effect caused by doped point defects in model materials[66],

$$
\begin{aligned}
f_{couple} = - \int d^3X(&\sigma_{11}{}^{loc}(X)Q_{11} + \sigma_{22}{}^{loc}(X)Q_{12} + \sigma_{33}{}^{loc}(X)Q_{12})P_1^2(X) \\
&+ (\sigma_{22}{}^{loc}(X)Q_{11} + \sigma_{11}{}^{loc}(X)Q_{12} + \sigma_{33}{}^{loc}(X)Q_{12})P_2^2(X) + (\sigma_{33}{}^{loc}(X)Q_{11} \\
&+ \sigma_{22}{}^{loc}(X)Q_{12} + \sigma_{22}{}^{loc}(X)Q_{12})P_3^2(X) + 2Q_{44}((\sigma_{12}{}^{loc}(X)P_1(X)P_2(X) \\
&+ \sigma_{13}{}^{loc}(X)P_1(X)P_3(X) + \sigma_{23}{}^{loc}(X)P_2(X)P_3(X)))
\end{aligned}
$$
$$(10)$$

where $\sigma_{ij}{}^{loc}(X), i,j = 1 - 3$ is the local stress fields created by the point defects due to the atomic size difference and does not change under cooling; $P_i(X)$ is the spontaneous polarization originated from the energy minimization of the total free energy and $Q_{ij}$ are the electrostrictive coefficients. The $f_{elas}$, and $f_{elec}$ are the elastic energy density, and the electrostatic energy density respectively. The elastic energy density can be expressed by:

$$f_{elas} = \frac{1}{2}c_{ijkl}e_{ij}e_{kl} = \frac{1}{2}c_{ijkl}(\varepsilon_{ij} - \varepsilon_{ij}^0)(\varepsilon_{kl} - \varepsilon_{kl}^0) \quad (11)$$

where $c_{ijkl}$ is the elastic stiffness tensor, $\varepsilon_{ij}$ the total strain, $\varepsilon^0{}_{kl}$ the electrostrictive stress-free strain, i.e., $\varepsilon^0{}_{kl} = Q_{ijkl}P_k P_l$. The electrostatic energy density can by expressed as:

$$f_{elec} = f_{dipole} + f_{depola} + f_{appl} \quad (12)$$

where $f_{dipole}$ is the dipole-dipole interaction caused by polarization, $f_{depola}$ is the depolarization energy density, and $f_{appl}$ is the energy density caused by applied electric field. The specific expressions of $f_{dipole}, f_{depola}$, and $f_{appl}$ are shown as follows:

$$f_{dipole} = -\frac{1}{2}E_i P_i \quad (13)$$

$$f_{depola} = -\frac{1}{2}E_{i,depola} \bar{P}_i \quad (14)$$

$$f_{appl} = -\frac{1}{2}E_{i,appl} \bar{P}_i \quad (15)$$

where $E_i$ represents the inhomogeneous electric field arising from dipole-dipole interactions, $E_{i,depola}$ denotes the mean depolarization field attributable to surface charges, the applied electric field is represented by a sine function and denoted as $E_{i,appl}$, and the spatial average value of the ith component of the polarization is expressed as

$$\bar{P}_i = \int P_i(X)/V \quad (16)$$

where $V$ represents the volume of the entire system.

$$E_{i,depola} = -\frac{1}{\varepsilon_i}\bar{P}_i \quad (17)$$

is an approximation for the average depolarization field caused by surface charges, where $\varepsilon_i = \varepsilon_0\varepsilon_b, \varepsilon_0$ is the vacuum dielectric constant and $\varepsilon_b$ denotes the background dielectric constant[67,68]. According to the time-dependent Ginzburg–Landau (TDGL) equation,

$$\frac{dP_i(X,t)}{dt} = -M\frac{\delta F}{\delta P_i(X,t)} \quad (18)$$

where $i = 1, 2, 3$, three order parameters ($P_1$, $P_2$, $P_3$) are defined to describe the domain structure. The change of the value of three order parameters can be used to distinguish Cubic ($C$) and Tetragonal ($T$), i.e., $P_C = P_0$ (0,0,0) describes $C$ phase, $P_T = P_0$ describes $T$ phase.

## Sample preparation

The raw compounds $SrCO_3$ (≥99.99%), $BaCO_3$ (≥99.5%), $Pb_3O_4$ (≥99.8%), $La_2O_3$ (≥99.95%), $Na_2CO_3$ (≥99.8%), and $Nb_2O_5$ (≥99.99%) were used as starting materials to synthesize $(Sr_{0.2}Ba_{0.2}Pb_{0.2}La_{0.2}Na_{0.2})Nb_2O_6$ via traditional solid-state method. After weighing, the powders were mixed with alcohol and zirconia beads and ball-milled for 4 h. Then, after drying, the mixture was calcined at 1160 °C for 3 h to obtain a single-phase TTB structure. The calcined powders were ball-milled again for 6 h, and then ground with 6.5 wt% PVA as a binder, pressed into pellets with a diameter of 13 mm, and then sintered at 1220 °C for 3 h to obtain the bulk ceramics.

## Structure characterizations

The crystal structures of the samples were characterized by an X-ray diffractometer (Bruker, D8 ADVANCE) with Cu Kα radiation (wavelength $\lambda = 1.5406$ Å). Temperature-dependent Raman spectra were tested on a Raman scattering spectrometer (Renishaw inVia). The microstructure was recorded by a S4800 Tabletop Microscope (Hitachi, Tokyo, Japan). The ferroelectric domain was characterized by commercial piezoelectric force microscopy (Jupiter XR, Oxford, UK). Transmission electron microscopy specimens for EDS patterns were meticulously prepared utilizing Ar$^+$ ion milling with the Gatan PIPS II, followed by a thin carbon coating to mitigate charging issues under the electron beam. Transmission electron microscopy specimens for obtaining HAADF imaging were prepared using Focused Ion Beam (FEI Scios 2 HiVac) with Ga$^+$ ions. JOEL JEM-F200 microscope was used to obtain TEM images. EDS patterns were obtained using a Titan Themis Z microscope. High-angle annular dark-field images at atomic scale were acquired using a probe-corrected transmission electron microscope (JOEL ARM200F). The convergence half-angle was 20.6 mrad, and the collection half-angle was between 54–220 mrad. Precise determination of atomic positions was achieved by StatSTEM software via fitting a spherical Gaussian with a specialized algorithm implemented in Matlab[69]. The formula

$$\text{ATAN2}\left((x_0 - x),(x_0 - y)\right) * \frac{180}{\text{PI()}} \quad (19)$$

$$\text{SQRT}\left((x_0 - x)^2 + (y_0 - y)^2\right) \quad (20)$$

were used to calculate the atomic displacement direction and magnitude. Atomic column intensities were extracted utilizing software developed by Lin et al.[70,71].

## Property measurements

To measure the electrical properties, circular Au electrodes (with diameter of 1 mm) were deposited onto the ceramic surface through a stainless-steel shadow mask. The $P$-$E$ loops and leakage current density were measured by a ferroelectric measuring system (aixACCT TF Analyzer 2000E) with the sample size of 0.07 mm (thickness) × 0.785 mm$^2$. The dielectric permittivity, loss tangent, and complex impedance were measured by a broad frequency/temperature dielectric spectrometer (Novocontrol GmbH, Concept 80). The charge-discharge performances of SBPLNN ceramics were evaluated using a commercial charge-discharge platform (CFD-003, Gogo Instruments Technology, Shanghai, China) with a discharge resistance of 300 Ω.

## Data availability

All data supporting this study and its findings are available within the article and its Supplementary Information. Any data deemed relevant are available from the corresponding author upon request.

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

## Acknowledgements

This work was supported by the National Natural Science Foundation of China (Grant No.12204503), the Key Research Program of the Chinese Academy of Sciences (Grant No. ZDRW-CN–2021–3-1-18), Shanghai Pujiang Program (No. 22PJD085) and Young Elite Scientists Sponsorship Program by China Association for Science and Technology (No. YESS20210265).

## Author contributions

Z.L. and G.W. conceived and designed this work. H.P. fabricated the samples, tested the properties, and processed the related data. T.W. and H.P. collected the TEM results. T.W., H.P., Z.F., Z.L. and F.X. analyzed the TEM results. D.W. and Y.H. performed the theory calculations. The manuscript was drafted by H.P. and revised by Z.L., G.W. and J.C. All authors participated in the data analysis and discussions.

## Competing interests

The authors declare no competing interests.
