## [Peer Review File · Nature Communications]

High-entropy relaxor ferroelectric ceramics for ultrahigh energy storageREVIEWER COMMENTS

Reviewer #1 (Remarks to the Author):

This manuscript describes a lead dielectric material by a high-entropy strategy to enhance energy storage performance. A giant recoverable energy density of 11.0 Jcm^{-3} and a high efficiency of 81.9% were achieved. The results are interesting and the manuscript can be published in NATURE COMMUNICATIONS after a major revision by addressing the following questions.

1. As the key point of the manuscript is a high-entropy strategy, the phase structure should be described in details. Because most of the studied ceramics show a strong relaxor behavior. This is common sense. The refinements of XRD or high-energy synchrotron XRD should be more believable.
2. In the results and discussion section, why the electric field intensity of charge and discharge tests are obviously lower than those of P-E measurements? Please give a reasonable explanation.
3. From the energy storage performances of SBPLNN ceramics, the efficiency decrease with the increase of electric field, and the explanation of related reasons is suggested. But the author proposed the strategy of a high-entropy including high permittivity and low dielectric loss.
4. Why choose Pb_3O_4 ? It is toxic, and not environmentally friendly.
5. The authors determined the atomic position through brightness and calculated the spontaneous polarization vector according to the B-position atomic displacement in each unit cell, the involved calculated software should be given in Experimental procedure.
6. The authors investigate the electrical heterogeneity by contrast spectroscopic plots of Z'' and M'' , however, The M'' axis needs to be rechecked.

Reviewer #2 (Remarks to the Author):

Authors have studied the novel materials with the high entropy approached design. However, I still feel that it can be improved and more suitable for the journal where routine work can be published with detailed investigations. Few comments to improve the manuscript for future communication. Discuss in details, at present it looks like observation and partially explained.

1. domain sizes gradually decrease as the number of element species increases. Moreover, the polarization directions progressively become disordered, thus delaying the polarization saturation process Explain properly and correlated with experimental data.
2. In this instance, we present a high-entropy tungsten bronze-type relaxor ferroelectric achieved through an equimolar-ratio element design, which realizes a giant recoverable energy density of 11.0 J/cm^{-3} and a high efficiency of 81.9%. The efficiency is not high as mentioned by the authors. Also, the mentioned recoverable energy density is not believable in such type of system and further please repeat the experiment provide data.
3. $(\text{Sr}_{0.2}\text{Ba}_{0.2}\text{Pb}_{0.2}\text{La}_{0.2}\text{Na}_{0.2}) \text{Nb}_2\text{O}_6$ selection of the material is not clear.
4. The atomic-scale micro-structural study confirms that the excellent comprehensive energy storage performance is attributed to the increased atomic-scale compositional heterogeneity from high configuration entropy, which modulates the relaxor features as well as induces lattice distortion, resulting in reduced polarization hysteresis and enhanced breakdown

endurance.

How high configuration entropy play such role for enhancing the property.

5. Tetragonal tungsten bronze (TTB) structures with the general formula $A_{12}A_{24}C_4B_{10}O_{30}$ attract extensive attention due to their complex structure and abundant dielectricity and ferroelectricity. Mention the draw back also.

6. Phase field method, explain properly and correlate with experimental data.

7. The overdamped (load resistance, $R = 100 \Omega$) discharge behavior of SBPLNN ceramics is illustrated in Fig. 2f. verify with other load resistance values and provide data.

8. The ΔS_{mix} rises to a high entropy of 1.61 R with five equimolar elements (Sr^{2+} , Ba^{2+} , Pb^{2+} , La^{3+} , and Na^{+}) introduced to the A sites, thus ultimately attaining thermodynamic stability by decreasing the free energy. What is the reason behind it and how to relate to enhance property.

Reviewer #3 (Remarks to the Author):

The authors demonstrate a high-entropy tungsten bronze-type relaxor ferroelectric achieved through an equimolar-ratio element design, which an unprecedented performance of recoverable energy density of $11.0 \text{ J}\cdot\text{cm}^{-3}$ and a high efficiency of 81.9%. This work shows that developing high-entropy relaxor ferroelectric tungsten bronze-type materials via equimolar-ratio element design is a promising approach to achieve high-performing energy storage capacitors. The paper is very-well written and the topic very interesting. As a minor correction, I would suggest that the authors at the end of the introduction clearly explain why they have decided to use the $Sr_{0.2}Ba_{0.2}Pb_{0.2}La_{0.2}Na_{0.2}Nb_2O_6$ composition in this study.

Dear Reviewers

Thanks very much for your efficient work and excellent comments on our manuscript (NCOMMS-24-07038). We have carefully considered and responded to your comments and made the revisions accordingly. The details are marked red in the revised paper. Meanwhile, we also responded to the comments point-by-point. We hope the revised manuscript will be suitable for publication in Nature Communications.

With regard to Reviewer 1's comments

General Overview

This manuscript describes a lead dielectric material by a high-entropy strategy to enhance energy storage performance. A giant recoverable energy density of $11.0 \text{ J}\cdot\text{cm}^{-3}$ and a high efficiency of 81.9% were achieved. The results are interesting and the manuscript can be published in *NATURE COMMUNICATIONS* after a major revision by addressing the following questions.

Overall Response:

We sincerely thank the reviewer for his/her positive comments and for recommending our manuscript for publication in *Nature Communications*.

Comment (1):

As the key point of the manuscript is a high-entropy strategy, the phase structure should be described in details. Because most of the studied ceramics show a strong relaxor behavior. This is common sense. The refinements of XRD or high-energy synchrotron XRD should be more believable.

Response (1):

Thanks for the reviewer's comment. As the reviewer indicated, detailed phase structure data is essential for the analysis of our novel high-entropy ceramics. Therefore, we performed the X-ray Rietveld refinement to provide detailed phase structure analysis and ensure that our experimental results are reliable. As seen in Figure R1, the XRD data of the SBPLNN high-entropy ceramics fits well with the Standard PDF Cards #45-0295, which proves a pure tetragonal tungsten bronze (TTB) structure with a *P4bm* tetragonal space group (lattice parameter: $a = 12.45\text{\AA}$, $b = 12.45\text{\AA}$, and $c =$

3.90Å).

We thank again for the reviewer's important comments. We have added the X-ray Rietveld refinement result to the revised Supplementary Materials and briefly discussed the phase structure in the main manuscript.

Figure R1. Rietveld refinement of the XRD data of SBPLNN high-entropy ceramics.

Comment (2):

In the results and discussion section, why the electric field intensity of charge and discharge tests are obviously lower than those of *P-E* measurements? Please give a reasonable explanation.

Response (2):

Thanks for the reviewer's comment. It should be noted that the same sample thickness (0.07mm) and electrode area (0.785 mm²) we used in the two different measurements. Therefore, **the reason for the difference in electric field intensity between the *P-E* measurements and charge and discharge tests should be attributed to the different types of voltage employed in the two tests.**

In the *P-E* measurements, we use **alternating current (AC) with a frequency of 10 Hz**, which can reflect the behavior of dielectric materials under a dynamic electric field. As seen in Figure R2, the period of the alternating current is 0.1 s, and it only withstands the maximum voltage at the moment of $t = 0.025$ s. Therefore, **the voltage duration of the maximum voltage is very short and for most of the time, the actual voltage applied on the sample is much smaller than the maximum value.** On the contrary, **the direct current (DC)** is used in charge and discharge tests, where the applied charged voltage maintains constant for ~2s. **The voltage duration is two orders of**

magnitude longer than that of the P - E measurements. Thus, it is more prone for the samples to breakdown. Consequently, the electric field endurance for charge and discharge tests is lower than that of P - E measurements.

To be specific, the charge and discharge test electric field ($500 \text{ kV}\cdot\text{cm}^{-1}$) of SBPLNN high-entropy ceramics in our work is about $2/3$ of that of P - E measurements ($753 \text{ kV}\cdot\text{cm}^{-1}$), which is similar to the situations of other relaxor ferroelectric ceramic investigations, as summarized in Table R1.

Figure R2. The voltage waveform diagram of P - E measurements (alternating current) and charge-discharge tests (direct current).

Table R1. The comparison of test electric fields of the P - E measurements and the charge-discharge testing of high performance relaxor ferroelectric ceramics reported in the literatures.

Components	P - E measurements ($\text{kV}\cdot\text{cm}^{-1}$)	charge and discharge tests ($\text{kV}\cdot\text{cm}^{-1}$)	Reference
$[(\text{K}_{0.2}\text{Na}_{0.8})_{0.8}\text{Li}_{0.08}\text{Ba}_{0.02}\text{Bi}_{0.1}]$ $(\text{Nb}_{0.68}\text{Sc}_{0.02}\text{Hf}_{0.08}\text{Zr}_{0.1}\text{Ta}_{0.08}\text{Sb}_{0.04})\text{O}_3$	740	240	Nat. Commun. 2022 ¹
$1/3\text{BaTiO}_3$ - $1/3\text{Bi}_{0.5}\text{Na}_{0.5}\text{TiO}_3$ - $1/3\text{NaNbO}_3$	550	240	Adv. Mater. 2022 ²
$0.848(\text{Na}_{0.52}\text{K}_{0.48})(\text{Sb}_{0.035}\text{Nb}_{0.965})\text{O}_3$ - 0.012SrZrO_3 - $0.14(\text{Bi}_{0.5}\text{Na}_{0.5})\text{ZrO}_3$	740	550	Adv. Mater. 2022 ³
$0.85\text{K}_{0.5}\text{Na}_{0.5}\text{NbO}_3$ - $0.15\text{Bi}(\text{Zn}_{2/3}\text{Ta}_{1/3})\text{O}_3$	600	420	Adv. Funct. Mater. 2021 ⁴
$0.75\text{Bi}_{0.35}\text{Na}_{0.35}\text{Sr}_{0.3}\text{TiO}_3$ - $0.15\text{Bi}(\text{Mg}_{2/3}\text{Ta}_{1/3})\text{O}_3$	560	400	Energy Environ. Sci. 2023 ⁵

0.90NaNbO ₃ -0.10BiFeO ₃	995	200	Energy Storage Mater. 2021 ⁶
Ba _{0.82} Bi _{0.12} TiO ₃	700	340	J. Am. Chem. Soc. 2023 ⁷
0.85(0.75Bi _{0.5} Na _{0.5} TiO ₃ -0.25BaTiO ₃)-0.15BaTiO ₃	660	350	J. Am. Chem. Soc. 2023 ⁸
Gd _{0.03} Ba _{0.47} Sr _{0.455} Sm _{0.02} Nb ₂ O ₆	660	320	Adv. Mater. 2023 ⁹
(Sr _{0.48} Na _{0.1} Bi _{0.26} Ca _{0.08} □ _{0.08})TiO ₃	440	300	Nano Energy 2023 ¹⁰
(Sr_{0.2}Ba_{0.2}Pb_{0.2}La_{0.2}Na_{0.2})Nb₂O₆	753	500	This work

Comment (3):

From the energy storage performances of SBPLNN ceramics, the efficiency decrease with the increase of electric field, and the explanation of related reasons is suggested. But the author proposed the strategy of a high-entropy including high permittivity and low dielectric loss.

Response (3):

Thanks for the reviewer's comment. As shown in Figure S3a (Supplementary Materials), SBPLNN high-entropy ceramics do realize a high permittivity (1064.62) at room temperature and 1 kHz. And the dielectric loss maintains a low value in a wide temperature and frequency range, for instance, as low as 0.0014 at room temperature and 1 kHz. Besides, as seen in Figure R3d, the achieved energy efficiency keeps higher than 85.0 % under $E \leq 450 \text{ kV}\cdot\text{cm}^{-1}$, which demonstrates the advantage of the proposed high-entropy strategy. With E further increasing to $753 \text{ kV}\cdot\text{cm}^{-1}$ the efficiency slightly decreases to 81.9%.

The decrease in efficiency is mainly attributed to the increased leakage current density with the increase in electric fields. The current responses at external electric fields mainly derive from PNRs switching (IPS), electric conductivity (IEC) and dielectric displacement (IDD). The electric conductivity can be calculated as: $I_{V_{\max}^+}^+ = I_{\text{EC}}^{V_{\max}^+} + I_{\text{DD}}^{V_{\max}^+}$, $I_{V_{\max}^-}^- = I_{\text{EC}}^{V_{\max}^-} + I_{\text{DD}}^{V_{\max}^-}$, $I_{\text{EC}}^{V_{\max}^+} = I_{\text{EC}}^{V_{\max}^-} = \frac{I_{V_{\max}^+}^+ + I_{V_{\max}^-}^-}{2}$. As seen in Figure R3a, the IEC increases with the increase in electric field. Therefore, we tested leakage current density, which also shows an inevitable increase with the increase in electric field and exhibits a faster growth rate under high electric fields (Figure R3b), thereby increasing the loss and remanent polarization, especially for high electric fields.

In addition, due to the polarization nonlinear response, the P_m increases much slower under higher electric field. Therefore, according to the calculation equation of energy storage parameters, the efficiency will slightly decrease during high E range.

Figure R3. a. I - E loop of SBPLNN ceramics and the schematic diagram of PNRs switching (IPS), electric conductivity (IEC) and dielectric displacement (IDD). b. Leakage current density as a function of the applied electric field. c. The variations of P_r , P_m and η till the maximum text electric field. d. W_{rec} , W_{tol} and η as a function of electric field.

Comment (4):

Why choose Pb_3O_4 ? It is toxic, and not environmentally friendly.

Response (4):

Thanks for the reviewer's comment. As mentioned in the introduction part, we designed the A-site high-entropy components based on the unfilled TTB structures, which attain electrical neutrality by filling 5/6 of the A-sites while leaving 1/6 A-site vacancies and therefore have the potential to produce additional entropy increment in the designed composition. Firstly, we choose the basic bivalent elements Sr^{2+} and Ba^{2+} from the classical unfilled TTB ceramic $(\text{Sr}_{0.5}\text{Ba}_{0.5})\text{Nb}_2\text{O}_6$. Then, considering the introduction of different valence ions, the commonly used modified ions La^{3+} and Na^+

are selected. In order to ensure electrical neutrality, another divalent ion is considered, like, Ca^{2+} and Pb^{2+} , which are also commonly doped in typical TTB compositions.

Then, we calculated the tolerance factor of $(\text{Sr}_{0.2}\text{Ba}_{0.2}\text{Pb}_{0.2}\text{La}_{0.2}\text{Na}_{0.2})\text{Nb}_2\text{O}_6$ and $(\text{Sr}_{0.2}\text{Ba}_{0.2}\text{Ca}_{0.2}\text{La}_{0.2}\text{Na}_{0.2})\text{Nb}_2\text{O}_6$, which are 0.957 and 0.947, respectively. The tolerance factor closes to 1 means a more stable crystal structure. This roughly indicates that Pb is more favorable for the formation of a pure TTB structure. Moreover, it is also demonstrated in existing literatures that a high content of Ca^{2+} can easily lead to the formation of a second phase¹¹. Actually, we have also synthesized the material based on the composition of $(\text{Sr}_{0.2}\text{Ba}_{0.2}\text{Ca}_{0.2}\text{La}_{0.2}\text{Na}_{0.2})\text{Nb}_2\text{O}_6$. In figure R4, the XRD of $(\text{Sr}_{0.2}\text{Ba}_{0.2}\text{Ca}_{0.2}\text{La}_{0.2}\text{Na}_{0.2})\text{Nb}_2\text{O}_6$ has an obvious impurity peak near 29°, which matches the CaNb_2O_6 second phase. On the contrary, **$(\text{Sr}_{0.2}\text{Ba}_{0.2}\text{Pb}_{0.2}\text{La}_{0.2}\text{Na}_{0.2})\text{Nb}_2\text{O}_6$ is crystallized in a pure TTB phase.** Forming a pure TTB structure is one of the important reasons for us to select potential elements.

Figure R4. The XRD patterns of $(\text{Sr}_{0.2}\text{Ba}_{0.2}\text{Pb}_{0.2}\text{La}_{0.2}\text{Na}_{0.2})\text{Nb}_2\text{O}_6$ and $(\text{Sr}_{0.2}\text{Ba}_{0.2}\text{Ca}_{0.2}\text{La}_{0.2}\text{Na}_{0.2})\text{Nb}_2\text{O}_6$ ceramics.

On the other hand, it is generally known that the polarization intensity of TTB ceramics is relatively lower compared with the perovskite structured ceramics, which may restrict the realization of super-high energy storage density. Thus, **Pb is selected for that the lone pair electronic $6s^2$ configuration of Pb^{2+} could strongly hybridize with O^{2-} 2p orbitals, giving rise to a high P_m , possibly contributing to a higher energy storage density of the high-entropy ceramics.**

Therefore, based on the above two considerations, we chose Pb^{2+} as one of the five A-site elements of our high-entropy ceramics.

Comment (5):

The authors determined the atomic position through brightness and calculated the spontaneous polarization vector according to the B-position atomic displacement in each unit cell, the involved calculated software should be given in Experimental procedure.

Response (5):

Thanks for the reviewer's comments. We use StatSTEM software to extract the atomic coordinates, and use EXCEL as a tool, with the formula $\text{ATAN2}((x_0-x), (x_0-y)) * 180/\text{PI}()$ and $\text{SQRT}((x_0-x)^2 + (y_0-y)^2)$ to calculate the atomic displacement directions and magnitude based on the atomic coordinates. We calculated the atomic displacement magnitude and directions of A sites atoms with respect to the corner atoms (Nb) and evaluate the lattice distortion of high-entropy ceramics.

We have added the corresponding calculated software to the experimental part of the revised manuscript.

Comment (6):

The authors investigate the electrical heterogeneity by contrast spectroscopic plots of Z'' and M'' , however, The M'' axis needs to be rechecked.

Response (6):

Thank the reviewer for the important reminder. We carefully checked the M'' axis according to the relevant references. We find that the way of marking the ordinate is slightly different. Some literatures use " $M''/\epsilon_0 (\text{F}^{-1} \text{cm} \cdot 10^8)$ "¹², and other literatures use " $M''/\epsilon_0 (\text{F}^{-1})$ "¹³, which mainly depends on whether " $\text{k}\Omega$ " or " $\text{k}\Omega \cdot \text{cm}^{-1}$ " is used as the coordinate axis unit of the impedance spectrum. For typical TTB ceramics, " $\text{k}\Omega$ " is used as the coordinate axis of the impedance spectra in previous literatures¹⁴⁻¹⁷. Thus, " $\text{k}\Omega$ " is also used as the coordinate axis of our SBPLNN high-entropy ceramics for a better comparison between our results and the literatures.

Therefore, in the investigations of the electrical heterogeneity by contrasting spectroscopic plots of Z'' and M'' , " $M''/\epsilon_0 (\text{F}^{-1})$ " is used as the coordinate axis unit of M'' . Thanks again for the reviewer's important reminder.

With regard to Reviewer 2's comments

General Overview

Authors have studied the novel materials with the high entropy approached design. However, I still feel that it can be improved and more suitable for the journal where routine work can be published with detailed investigations. Few comments to improve the manuscript for future communication. Discuss in details, at present it looks like observation and partially explained.

Overall Response:

We sincerely thank the reviewer for your positive recognition of our achievements and important comments/suggestions for further improving our work. We have listed answers and solutions to your comments. Please check them.

Comment (1):

Domain sizes gradually decrease as the number of element species increases. Moreover, the polarization directions progressively become disordered, thus delaying the polarization saturation process. Explain properly and correlated with experimental data.

Response (1):

Thanks for the reviewer's comments. Regulating the local compositional inhomogeneity is a common approach to tuning the relaxor properties. For our high-entropy strategies, with the increase in element species, the local component heterogeneity is enhanced, which can be quantitatively evaluated by the increased configuration entropy. The equimolar foreign atoms with unmatched atomic size, mass, and electronegativity act as point defects of the identical proportion, gradually destroying the ferroelectric long-range order into PNRs. Therefore, with the highly atomic disorder that induces local structure disorder and polar fluctuations, thus the domain sizes gradually decrease as the number of element species increases. The polarization direction of ultra-small PNRs thus becomes more disordered, and the enhancement of relaxor characteristics effectively delays the saturation polarization, enabling the achievement of high P_m under a higher electric field. **We have added more explanation about this to improve the readability of this part.**

According to the reviewer's comment, we have improved the correlation between phase field simulation and experimental result and supplemented the discussion on the experimental

results of piezo-response force microscopy (PFM) and transmission electron microscopy (TEM) between our 5-ary SBPLNN high-entropy ceramics and 2-ary $\text{Sr}_{0.5}\text{Ba}_{0.5}\text{Nb}_2\text{O}_6$ (SBN) and $\text{Pb}_4\text{Na}_2\text{Nb}_{10}\text{O}_{30}$ (PNN) low-entropy ceramics. As seen in Figure R5, different from the large size initial domain of PNN and SBN¹⁸, SBPLNN ceramics show ultrasmall spot-like nanodomains. Moreover, SBN and PNN display more evident piezoelectric signal after 15 V and 40 V voltage poling, respectively. The piezoelectric signal still can be clearly observed in SBN¹⁷ and PNN after 15 min relaxation duration. However, under a high electric field of 40 V, SBPLNN ceramics present a much weaker piezoelectric response signal, indicating a delayed polarization response to applied electric field compared with low-entropy ceramics. The piezoelectric response signal completely disappears after 15 min, revealing the enhanced domain mobility.

Figure R5. Out-of-plane PFM phase images of a₁. PNN, b₁. SBN¹⁸, and c₁. SBPLNN. Phase images of a₂-a₄. PNN, b₂-b₄. SBN¹⁷, and c₂-c₄. SBPLNN upon application with a voltage (15 V for SBN, 40 V for PNN and SBPLNN) and different relaxation durations.

Furthermore, the domain structures of PNN, SBN, and SBPLNN were directly observed by TEM. As shown in Figure R6a-b, PNN exhibits large strip domains with diameters of ~200 nm and SBN possess a large domain size of about 42 nm. For SBPLNN, the long-range ferroelectric order is completely disrupted into spot-like PNRs and no obvious large-size domain structure can be observed. **The domain structure changes measured by TEM are consistent with the results of PFM, which strongly correlates and demonstrates the phase field simulation results.**

In conclusion, these results are consistent with the analysis of the phase field simulation. Relevant discussions and experimental results have been added to the revised manuscript. The consistency between theoretical analysis and experimental results confirms the feasibility of our equimolar-ratio high-entropy strategy.

Figure R6. The HRTEM images of the domain morphology of a. PNN, b.SBN¹⁷, and C. SBPLNN ceramics.

Comment (2):

In this instance, we present a high-entropy tungsten bronze-type relaxor ferroelectric achieved through an equimolar-ratio element design, which realizes a giant recoverable energy density of 11.0 J/cm⁻³ and a high efficiency of 81.9%. The efficiency is not high as mentioned by the authors. Also, the mentioned recoverable energy density is not believable in such type of system and further please repeat the experiment provide data.

Response (2):

We thank the reviewer very much for the important comment. In recent years, investigations with

regards on the energy storage performance of TTB ceramics based on various strategies have been reported from time to time. Generally, the relaxor behavior can be easily achieved in TTB ceramics via constructing the random distribution of site atoms, which is contemplated to be one of the main structural origins of the enhanced relaxor behavior. Moreover, TTB structure have two nonequivalent A sites (A1 and A2 sites) and diversified kinds of structure (unfilled, filled, and fully filled), which endow the TTB a great degree of flexibility in composition design and energy storage performance realization. Therefore, great progresses on the energy storage study of TTB have been achieved recently, including, W_{rec} of **5.9 J·cm⁻³** and an η of 85% in $\text{Sr}_{0.425}\text{La}_{0.1}\square_{0.05}\text{Ba}_{0.425}\text{Nb}_{1.4}\text{Ta}_{0.6}\text{O}_6$ (Haonan Peng et al. *Adv. Sci.* **2023**, 10, 2300227)¹⁷, W_{rec} of **6.2 J·cm⁻³** and η of 85% in $\text{Sr}_{0.5}\text{Ba}_{0.47}\text{Gd}_{0.02}\text{Nb}_{1.8}\text{Ta}_{0.2}\text{O}_6$ (Bian Yang et al. *Energy Stor. Mater.* **2023**, 55, 763-772)¹⁹, and W_{rec} of **9 J·cm⁻³** and η of 84% in $\text{Gd}_{0.03}\text{Ba}_{0.47}\text{Sr}_{0.455}\text{Sm}_{0.02}\text{Nb}_2\text{O}_6$ (Yangfei Gao et al. *Adv. Mater.* **2023**, 2310559)⁹. **Because of the flexible structures and abundant property tunability, the TTB family actually possesses a great potential for the realization of high energy storage performance. This lays the foundation for the attainment of such a high energy storage property in our current work.**

According to the reviewer's suggestions, we repeated the *P-E* loop measurements and provided corresponding parameters by testing another 7 SBPLNN samples, including P_m , P_r , E , W_{rec} , W_{tot} , and η , as shown in Figure R7 and Table R2. It can be seen that each parameter exhibits acceptable minor fluctuations. Moreover, **the energy storage density calculated by 7 repeated experiments concerned by the reviewers fluctuates slightly around 11.0 J·cm⁻³**. Therefore, the reliability of the recoverable energy density of our tungsten bronze structured composition is well confirmed.

As for the energy storage efficiency, from the data in Table R2, the calculated energy storage efficiency slightly fluctuates between 80.1% and 84.8% among the 8 test results. Therefore, the efficiency of 81.9 % in our manuscript is also reliable. Besides, as seen in Figure 2b (in the main text), efficiency higher than 85 % can be achieved under $E \leq 450 \text{ kV}\cdot\text{cm}^{-1}$. With the applied electric field increasing to $753 \text{ kV}\cdot\text{cm}^{-1}$, only a slight decrease to 81.9% is detected. As also responded to reviewer 1' comment 3, considering the increased leakage current under such an ultra-high electric field of $753 \text{ kV}\cdot\text{cm}^{-1}$ in our study, the efficiency of 81.9 % is actually can be considered as a high value for TTB ceramics, compared with the existing literatures^{9,17,19}.

Figure R7. The P - E loops and energy storage performance of 8 SBPLNN samples.

Table R2. The P_r , P_m , E_b , W_{rec} , W_{total} , and η of 8 SBPLNN samples.

Sample	P_m (μC·cm ⁻²)	P_r (μC·cm ⁻²)	E (kV·cm ⁻¹)	W_{rec} (J·cm ⁻³)	W_{tol} (J·cm ⁻³)	η (%)
1	43.45	3.42	733	10.4	12.5	83.1
2	45.27	3.80	768	10.6	13.3	80.1
3	45.26	3.92	774	11.2	13.7	81.9
4	43.71	1.89	794	11.1	13.1	84.7
5	45.38	3.24	733	10.7	12.7	83.6
6	42.78	2.23	820	10.8	12.8	84.8
7	44.45	3.88	754	10.7	13.2	81.2
Manuscript	45.90	2.84	753	11.0	13.4	81.9

Comment (3):

($\text{Sr}_{0.2}\text{Ba}_{0.2}\text{Pb}_{0.2}\text{La}_{0.2}\text{Na}_{0.2}\text{Nb}_2\text{O}_6$) selection of the material is not clear.

Response (3):

Thank the reviewer for the important comment. Firstly, unfilled TTB attain electrical neutrality by filling 5/6 of the A-sites while leaving 1/6 A-site vacancies, which allows the disordered distribution of cations and vacancies, showing the potential to produce additional entropy increment. Therefore, **we choose the unfilled structure as the prototype for high-entropy design**. Secondly, considering as many different valence ions as possible, divalent ions, trivalent ions and monovalent ions are introduced to one TTB cell on the basis of maintaining the 1/6 vacancy of the unfilled structures. Therefore, **we choose the common elements Sr^{2+} and Ba^{2+} from the classical unfilled TTB ceramic $(\text{Sr}_{0.5}\text{Ba}_{0.5})\text{Nb}_2\text{O}_6$. Subsequently, the trivalent modified ion La^{3+} was chosen, meanwhile, the common monovalent Na^+ was introduced to maintain electrical neutrality**. Furthermore, it is generally known that the polarization intensity of TTB is relatively lower compared with the perovskite structured materials. **The lone pair electronic $6s^2$ configuration of Pb^{2+} could strongly hybridize with O^{2-} 2p orbitals, giving rise to a high P_m , contributing to the potential achievement of higher energy storage performance**. Therefore, we select Pb^{2+} as one of the five elements for our high-entropy design.

We have briefly added necessary discussions with regards to the composition selection to the end of the introduction of our revised manuscript.

Comment (4):

The atomic-scale micro-structural study confirms that the excellent comprehensive energy storage performance is attributed to the increased atomic-scale compositional heterogeneity from high configuration entropy, which modulates the relaxor features as well as induces lattice distortion, resulting in reduced polarization hysteresis and enhanced breakdown endurance. How high configuration entropy play such role for enhancing the property.

Response (4):

Thanks for the reviewer's comments. Due to the different coordination environment and space size of A1 and A2 sites, the TTB structures usually possess inherent atomic A1 or A2 site selectivity according to the ionic radius. Generally speaking, larger atoms are prone to enter A2-sites, while,

smaller ones tend to occupy A1 sites. For the SBPLNN high-entropy ceramics, we introduce five ions with different valence states and radii into the A-site of TTB structures in equal proportion to cause the maximum configuration entropy ($1.61R$). Subsequently, we confirmed by HAADF-STEM that **the high configuration entropy breaks the atomic site selectivity that depends on the difference in ion radius, which in turn causes an enhanced local compositional inhomogeneity**. As shown in Figure 3c (the main text), all five A-site elements enter the A1 and A2 sites and the ratio is near the ideal ratio of 1:2. This high atomic disorder causes local polarity and structural fluctuations, thus regulating the relaxor characteristics. As displayed in Figure R5 and Figure R6, the PFM and TEM results indicated that the long-range ferroelectric order in 2-ary low-entropy ceramics is disrupted, polar nanoregions with enhanced domain mobility and lower switching barriers are generated in SBPLNN high-entropy ceramics, resulting in slim P - E loops with high P_m and low P_r .

On the other hand, the five highly disordered ions of high-entropy SBPLNN ceramics make the adjacent atoms in the lattice have different radius, mass and electronegativity, which cause random deviation from their perfect sites and strong non-periodic lattice distortion. As shown in Figure 4a-d (the main text), the direction of the atomic displacement randomly distributed from -180° to 180° , and presented a large atomic displacement. The increased lattice distortion can increase the crystalline energy to a level that cannot be compensated by the shrinking of grain surface areas and, thus, inhibit grain coarsening. Meanwhile, activated lattices and decreased sintering temperature would be triggered by a high-entropy design with the introduction of multiple elements, which would be in favor of grain refinement. Furthermore, strong lattice distortion generally increases the probability of collision and scattering of electrons with twisted lattice atoms. **Thus, the electrical insulation performance (impedance increases, leakage current decreases) is improved, and the electrical uniformity is enhanced**. Finally, ultra-high breakdown strength is achieved in our SBPLNN high-entropy ceramics.

In conclusion, the equimolar-ratio high-entropy strategy enhances the local component heterogeneity, effectively regulates the relaxor characteristics, which reduces the residual polarization intensity and delays the saturation polarization. At the same time, the strong non-periodic lattice distortion refines the grains, improves the insulation performance and enhances the electrical homogeneity, and achieves ultra-high breakdown strength. Finally, excellent energy storage performance of $11.0 \text{ J}\cdot\text{cm}^{-3}$ and 81.9% are obtained. We have clearly explained the effect of the high

entropy design in enhancing the properties.

Additionally, we also calculate the ΔS_{config} of the reported advanced bulk ceramics and perform a comparison of their W_{rec} as well as the ΔS_{config} in Figure R8. Regarding the whole, the W_{rec} displays an increment trend with the increase of configurational entropy, and our SBPLNN ceramics occupy the highest position among these reported materials with different entropy values. The above analysis and discussion fully illustrate that high configuration entropy plays a positive role for enhancing the property.

Figure R8. Comparison of the W_{rec} of reported advanced bulk ceramics with various ΔS_{config} summarized by this work.

Comment (5):

Tetragonal tungsten bronze (TTB) structures with the general formula $(A1)_2(A2)_4C_4B_{10}O_{30}$ attract extensive attention due to their complex structure and abundant dielectricity and ferroelectricity. Mention the drawback also.

Response (5):

Thanks for the reviewer's comments. The TTB ceramics indeed have two main drawbacks. **The lower surface energy of the (001) facet of TTB ceramics induces faster grain growth than other directions during sintering progress. Therefore, the grains tend to grow abnormally into rod-shaped, large-size grains, resulting in a relative low dielectric strength. Meanwhile, the**

polarization of TTB ceramics is generally lower than that of perovskite structured ferroelectric ceramics, which may restrict the realization of super-high energy storage density. We have added the above drawbacks in the 12th line of the third paragraph of the introduction part in our revised manuscript.

However, in our work, the two main drawbacks of TTB ceramics can be overcome through constructing high entropy. Because of our equimolar-ratio high-entropy strategy, the strong non-periodic lattice distortion refines the grains, improves the insulation performance and enhances the electrical homogeneity, and therefore achieves an ultra-high dielectric strength of $\sim 753 \text{ kV}\cdot\text{cm}^{-1}$. The high configuration entropy enhances local compositional inhomogeneity and induces polar nanoregions with enhanced domain mobility and lower switching barriers in SBPLNN high-entropy ceramics, resulting in slim P - E loops with high P_m and low P_r . At the same time, the introduction of Pb^{2+} also enhances the polarization strength. Finally, SBPLNN high-entropy ceramics obtain a large ΔP of $43 \mu\text{C}\cdot\text{cm}^{-2}$ under an ultra-high electric field.

Comment (6):

Phase field method, explain properly and correlate with experimental data.

Response (6):

Thanks for the reviewer's comments. In the phase field simulation part, we mainly simulated and discussed the following two aspects of evolution:

1. Changes of domain structure with increased element species and their responses under applied electric fields. As the reviewer has also concerned the explanation of this part in the comment 1, we have discussed in detail the explanation and correlated it with experimental data in the response to comment 1. We will only discuss another aspect here in order to avoid lengthy and jumbled problem.

2. The stable domain structure evolution of 5-ary high-entropy ceramics in a wide temperature range.

As one of the main arguments in the manuscript, the relaxor behavior of 5-ary high-entropy ceramics gets enhanced, and the long-range ferroelectric domains are destroyed into ultra-small PNRs. Generally, relaxor ferroelectrics exhibit a diffused phase transition and possess a relatively flat dielectric spectrum. Therefore, the dielectric relaxation takes place over a broad temperature range,

and the dielectric constant changes little with temperature variation. **Consequently, the ultra-small PNRs can exist in a wide dielectric relaxation temperature range, thus exhibiting temperature-insensitive macroscopic polarization behavior.** Meanwhile, the presence of highly-active PNRs possess lower polarization switching barriers, thus ensuring slim P - E loops over a broad temperature range. We have modified the explanation of this part.

We have also further correlated this simulation with related experiment results, including property measurements and structural analysis. In Figure S3a (supplementary materials), the dielectric peak drops to about $-100\text{ }^{\circ}\text{C}$, and as the temperature increases from $-120\text{ }^{\circ}\text{C}$ to $120\text{ }^{\circ}\text{C}$, SBPLNN ceramics experience transition from the relaxor phase to the superparaelectric phase, and finally into the paraelectric region. Therefore, as seen in Fig. S8 (supplementary materials), the size and content of PNRs keeps overall stable while slightly decreases, making P_m slightly decrease from $36.43\text{ }\mu\text{C}\cdot\text{cm}^{-2}$ to $31.94\text{ }\mu\text{C}\cdot\text{cm}^{-2}$. Moreover, PNRs maintain high activity in a wide temperature range, it responds quickly to the applied electric field and demonstrate facilitated polarization rotation, thus ensuring slim P - E loops with high P_m and low P_r . Finally, the narrow feature of P - E loops stays unchanged, resulting in a stable high $W_{\text{rec}} \sim 5.50 \pm 0.12\text{ J}\cdot\text{cm}^{-3}$ and $\eta \sim 86.54 \pm 1.51\%$ throughout the broad testing temperature range. Temperature-dependent XRD and Raman spectroscopy are also employed to investigate the thermal structural evolution. The positions and numbers of diffraction peaks remain unaltered with a temperature change from $-120\text{ }^{\circ}\text{C}$ to $290\text{ }^{\circ}\text{C}$ (Figure S6. in supplementary materials), indicating the structure stability of SBPLNN ceramics. As depicted in Figure S7 (supplementary materials), across the broad temperature range of $-180\text{ }^{\circ}\text{C}$ to $300\text{ }^{\circ}\text{C}$, each Raman mode exhibits a diffused and broadened characteristic, which signifies the potent relaxor nature of SBPLNN ceramics. External vibration modes (below 200 cm^{-1}) indicate strong scattering intensity and complex peak shapes, which are associated with the distinct charges and radii between A site atoms of SBPLNN ceramics. This cation disorder contributes to the local random electric fields and induces the formation of PNRs in a wide temperature range.

We have made necessary revisions based on the above response, which have been marked in red in the revised manuscript.

Comment (7):

The overdamped (load resistance, $R=100\text{ }\Omega$) discharge behavior of SBPLNN ceramics is

illustrated in Fig. 2f. verify with other load resistance values and provide data.

Response (7):

According to the reviewer's comment, we provide the overdamped discharge behavior under a load resistance of 200 Ω and 300 Ω . At the same time, we also re-tested the over-damped discharge performance under same respective load resistance. As shown in figure R9, under the same electric field of 500 $\text{kV}\cdot\text{cm}^{-1}$, as the load increases (from 100 Ω to 200 Ω and 300 Ω), the peak current gradually decreases, and the discharge period $t_{0.9}$ increases from 26.6 ns to 38.2 and 51.2 ns. In addition, the W_{dis} rises slightly, which may be attributed to the slower discharge speed reducing the loss during the domain flipping process.

Over-damped discharge tests under different loads and repeated experiments confirmed the reliability of the data and proved the excellent discharge performance of our high-entropy ceramics. **Since the discharge test under high loads is typically employed in the literatures, to offer a better comparison, we replace the original data with the discharging under 300 Ω in the revised manuscript.**

Figure R9 a. The overdamped discharge performance under a load resistance of 100 Ω , 200 Ω , and 300 Ω . **b.** The repeated experiment results under same respective load resistances.

Comment (8):

The ΔS_{mix} rises to a high entropy of 1.61 R with five equimolar elements (Sr^{2+} , Ba^{2+} , Pb^{2+} , La^{3+} , and Na^+) introduced to the A sites, thus ultimately attaining thermodynamic stability by decreasing the free energy. What is the reason behind it and how to relate to enhance property.

Response (8):

Thanks for the reviewer's comments. In the realm of thermodynamics, entropy maintains a close relationship with Gibbs free energy (ΔG), a link corroborated by the Gibbs–Helmholtz equation: $\Delta G_{\text{mix}} = \Delta H_{\text{mix}} - T\Delta S_{\text{mix}}$, where ΔH_{mix} signifies enthalpy and T represents the absolute temperature of the system. The configurational entropy ($T\Delta S_{\text{mix}}$) increases with increasing element species, when the increase of entropy is larger than that of enthalpy ($T\Delta S_{\text{mix}}$ exceeds ΔH_{mix}), leading to the mix Gibbs free energy $\Delta G_{\text{mix}} < 0$, and the change of energy manifests as the entropy-driven structural stabilization effect, where a high-entropy structure can be stabilized^{20, 21}. Such a high-entropy effect expands the solution bounds of conventional compounds and provides a feasible route for the formation of random solid solution compounds.

The above discussion on the free energy is mainly based on the thermodynamic point of view to discuss the promoting effect of high configuration entropy on the formation of stable single-phase structure, which is not directly related with the properties. While, the enhancement of energy storage performance originates from the enhanced relaxor behavior and increased breakdown strength, which originates from the high-entropy induced local compositional heterogeneity and strong non-periodic lattice distortion.

With regard to Reviewer 3's comments

General Overview

The authors demonstrate a high-entropy tungsten bronze-type relaxor ferroelectric achieved through an equimolar-ratio element design, which an unprecedented performance of recoverable energy density of $11.0 \text{ J}\cdot\text{cm}^{-3}$ and a high efficiency of 81.9%. This work shows that developing high-entropy relaxor ferroelectric tungsten bronze-type materials via equimolar-ratio element design is a promising approach to achieve high-performing energy storage capacitors. The paper is very-well written and the topic very interesting. As a minor correction, I would suggest that the authors at the end of the introduction clearly explain why they have decided to use the $(\text{Sr}_{0.2}\text{Ba}_{0.2}\text{Pb}_{0.2}\text{La}_{0.2}\text{Na}_{0.2})\text{Nb}_2\text{O}_6$ composition in this study.

Response:

We sincerely thank the reviewer for his/her positive comments and for recommending our manuscript for publication in *Nature Communications*. The reasons for choosing the $(\text{Sr}_{0.2}\text{Ba}_{0.2}\text{Pb}_{0.2}\text{La}_{0.2}\text{Na}_{0.2})\text{Nb}_2\text{O}_6$ composition are as follows:

As mentioned in introduction, unfilled TTB attain electrical neutrality by filling 5/6 of the A-sites while leaving 1/6 A-site vacancies, vacancies can be randomly distributed in both the A1-sites and A2-sites, leading to chaotic states of the A-site vacancies and cations, showing the potential to generate further entropy increment with higher atomic disorder and compositional heterogeneity. **Therefore, we choose the unfilled structure as the prototype for high-entropy design.** Then, under the consideration of adopting as many different valence ions as possible, divalent ions, trivalent ions and monovalent ions are introduced to one TTB cell on the basis of maintaining the 1/6 vacancy of the unfilled structures to maintain electrical neutrality. Therefore, **we choose the basic bivalent elements Sr^{2+} and Ba^{2+} from the classical unfilled TTB ceramic $(\text{Sr}_{0.5}\text{Ba}_{0.5})\text{Nb}_2\text{O}_6$.** Subsequently, **the trivalent modified ion La^{3+} is chosen, meanwhile, the common monovalent Na^+ is introduced to maintain electrical neutrality.** Furthermore, it is generally known that the polarization intensity of TTB ceramics is relatively lower compared with the perovskite structured materials. **The lone pair electronic $6s^2$ configuration of Pb^{2+} could strongly hybridize with O^{2-} 2p orbitals, giving rise to a high P_m , contributing to the potential achievement of higher energy storage performance.**

Therefore, we select Pb^{2+} as one of the five elements for our high-entropy design.

We have briefly added necessary discussions with regards to the composition selection to the end of the introduction of our revised manuscript.

Reference

1. Chen L, *et al.* Giant energy-storage density with ultrahigh efficiency in lead-free relaxors via high-entropy design. *Nat. Commun.* **13**, 3089, (2022).
2. Chen L, *et al.* Local Diverse Polarization Optimized Comprehensive Energy-Storage Performance in Lead-Free Superparaelectrics. *Adv. Mater.* **33**, 2205787, (2022).
3. Xie A, *et al.* Supercritical Relaxor Nanograined Ferroelectrics for Ultrahigh-Energy-Storage Capacitors. *Adv. Mater.* **34**, 2204356, (2022).
4. Li D, *et al.* Improved Energy Storage Properties Achieved in (K, Na)NbO₃-Based Relaxor Ferroelectric Ceramics via a Combinatorial Optimization Strategy. *Adv. Funct. Mater.* **32**, (2021).
5. Li D, *et al.* A high-temperature performing and near-zero energy loss lead-free ceramic capacitor. *Energy Environ. Sci.* **16**, 4511-4521, (2023).
6. Jiang J, *et al.* Ultrahigh energy storage density in lead-free relaxor antiferroelectric ceramics via domain engineering. *Energy Storage Mater.* **43**, 383-390, (2021).
7. Sun Z, *et al.* Superior Capacitive Energy-Storage Performance in Pb-Free Relaxors with a Simple Chemical Composition. *J. Am. Chem. Soc.* **145**, 6194-6202, (2023).
8. Liu H, *et al.* Chemical Design of Pb-Free Relaxors for Giant Capacitive Energy Storage. *J. Am. Chem. Soc.* **145**, 11764-11772, (2023).
9. Gao Y, *et al.* Ultrahigh Energy Storage in Tungsten Bronze Dielectric Ceramics Through a Weakly Coupled Relaxor Design. *Adv. Mater.* **36**, 2310559, (2023).
10. Liu L, *et al.* Multi-scale collaborative optimization of SrTiO₃-based energy storage ceramics with high performance and excellent stability. *Nano Energy* **109**, 108275, (2023).
11. Hai L, *et al.* Effect of the ratio of Sr to Ba at A2 sites on the dielectric and ferroelectric properties of tetragonal tungsten bronze Ca_{0.4}Sr_xBa_{0.6-x}Nb₂O₆ ceramics. *Journal of Materials Science: Materials in Electronics* **33**, 14655-14662, (2022).
12. Lu Z, *et al.* Superior energy density through tailored dopant strategies in multilayer ceramic capacitors. *Energy Environ. Sci.* **13**, 2938-2948, (2020).
13. Wang G, *et al.* Ultrahigh energy storage density lead-free multilayers by controlled electrical homogeneity. *Energy Environ. Sci.* **12**, 582-588, (2019).
14. Zhang X, *et al.* Simultaneously Realizing Superior Energy Storage Properties and Outstanding Charge-Discharge Performances in Tungsten Bronze-Based Ceramic for Capacitor Applications. *Inorg. Chem.* **60**, 6559-6568, (2021).
15. Wang H, *et al.* Pb/Bi-free Tungsten Bronze-Based Relaxor Ferroelectric Ceramics with Remarkable Energy Storage Performance. *ACS Appl. Energy Mater.* **4**, 9066-9076, (2021).
16. Luo C, *et al.* Promoting Energy Storage Performance of Sr_{0.7}Ba_{0.3}Nb₂O₆ Tetragonal Tungsten Bronze Ceramic by a Two-Step Sintering Technique. *ACS Appl. Electron. Mater.* **4**, 452-460, (2021).
17. Peng H, *et al.* Superior Energy Density Achieved in Unfilled Tungsten Bronze Ferroelectrics via Multiscale Regulation Strategy. *Adv. Sci.* **10**, 2300227, (2023).
18. Huang CJ, *et al.* Variation of ferroelectric hysteresis loop with temperature in (Sr_xBa_{1-x})Nb₂O₆ unfilled tungsten bronze ceramics. *J. Mater.* **1**, 146-152, (2015).
19. Yang B, *et al.* Remarkable energy storage performances of tungsten bronze Sr_{0.53}Ba_{0.47}Nb₂O₆-based lead-free relaxor ferroelectric for high-temperature capacitors application. *Energy Stor. Mater.* **55**, 763-772,

(2023).

20. Jiang B, *et al.* High-entropy-stabilized chalcogenides with high thermoelectric performance. *Science* **371**, 830-834, (2021).
21. Yang B, *et al.* High-entropy design for dielectric materials: Status, challenges, and beyond. *J. Appl. Phys.* **133**, 110904, (2023).

REVIEWERS' COMMENTS

Reviewer #1 (Remarks to the Author):

The revised version well addressed all the questions I raised. The responses are convincing. The manuscript can be accepted for publication.

Reviewer #2 (Remarks to the Author):

It can be accepted

Reviewer #3 (Remarks to the Author):

The authors addressed my major concern. Therefore, I can recommend this manuscript for publication.

Dear Reviewers

Thanks again for your efficient work and constructive comments on our manuscript (NCOMMS-24-07038A). And we also sincerely thank the reviewers for the recommendation of publication. The point-to-point responses are listed below.

With regard to Reviewer 1's comments

Comment:

The revised version well addressed all the questions I raised. The responses are convincing. The manuscript can be accepted for publication.

Response:

We sincerely thank the reviewer for the recognition of our revision to the manuscript and for recommending our manuscript for publication in *Nature Communications*.

With regard to Reviewer 2's comments

Comment:

It can be accepted.

Response:

We are grateful for your thoughtful comments on the manuscript and for your acceptance recommendation.

With regard to Reviewer 3's comments

Comment:

The authors addressed my major concern. Therefore, I can recommend this manuscript for publication.

Response:

We sincerely appreciate your valuable review and recommendation for the acceptance of our manuscript.